# Computational and Experimental Evidence for Templated Macrocyclization: The Role of a Hydrogen Bond Network in the Quantitative Dimerization of 24-Atom Macrocycles

**DOI:** 10.3390/molecules28031144

**Published:** 2023-01-23

**Authors:** Alexander J. Menke, Nicholas C. Henderson, Lola C. Kouretas, Anne N. Estenson, Benjamin G. Janesko, Eric E. Simanek

**Affiliations:** Department of Chemistry & Biochemistry, Texas Christian University, Fort Worth, TX 76109, USA

**Keywords:** macrocycle, preorganization, hydrogen bonding, triazine, hydrazone, computation

## Abstract

In the absence of preorganization, macrocyclization reactions are often plagued by oligomeric and polymeric side products. Here, a network of hydrogen bonds was identified as the basis for quantitative yields of macrocycles derived from the dimerization of monomers. Oligomers and polymers were not observed. Macrocyclization, the result of the formation of two hydrazones, was hypothesized to proceed in two steps. After condensation to yield the monohydrazone, a network of hydrogen bonds formed to preorganize the terminal acetal and hydrazine groups for cyclization. Experimental evidence for preorganization derived from macrocycles and acyclic models. Solution NMR spectroscopy and single-crystal X-ray diffraction revealed that the macrocycles isolated from the cyclization reaction were protonated twice. These protons contributed to an intramolecular network of hydrogen bonds that engaged distant carbonyl groups to realize a long-range order. DFT calculations showed that this network of hydrogen bonds contributed 8.7 kcal/mol to stability. Acyclic models recapitulated this network in solution. Condensation of an acetal and a triazinyl hydrazine, which adopted a number of conformational isomers, yielded a hydrazone that adopted a favored rotamer conformation in solution. The critical hydrogen-bonded proton was also evident. DFT calculations of acyclic models showed that the rotamers were isoenergetic when deprotonated. Upon protonation, however, energies diverged with one low-energy rotamer adopting the conformation observed in the macrocycle. This conformation anchored the network of hydrogen bonds of the intermediate. Computation revealed that the hydrogen-bonded network in the acyclic intermediate contributed up to 14 kcal/mol of stability and preorganized the acetal and hydrazine for cyclization.

## 1. Introduction

The cyclic, undecapeptide cyclosporin, is an archetype for the design of bioactive macrocyclic molecules. Comprising a ring of 33 atoms, cyclosporin serves as a foundation for building out the so-called BRo5 (beyond Lipinski’s Rule of 5) principles [1,2,3] that define drug-like characteristics for large molecules [4]. These design principles address the roles of conformational flexibility (so-called chameleonicity) [5], *N*-alkylation [6], and solubility [7,8]. Cyclosporin also serves to anchor new computational modeling strategies [9]. Given the current interests in these topics, cyclosporin’s historic role in shaping synthetic strategy receives less attention. It served as a model for evaluating methodology as well as identifying sites for cyclization [10]. In addition, and more germane to this work, the synthesis of cyclosporin points to the role that hydrogen bonding networks can play in retrosynthetic analysis. Although executed at high dilution (10^−3^–10^−4^ M), the high yield (62%) obtained for cyclization was attributed to preorganization of the linear peptide into a product-like conformation.

In facilitating macrocycle synthesis, preorganization serves as a powerful strategy for influencing product distributions [11,12]. Oftentimes, this distribution can be defined as discrete versus polymeric products. An early example of the use of hydrogen bonding to promote macrocyclization over polymerization involved the condensation of a dialdehyde with a diamine. Intramolecular hydrogen bonds preorganized the aldehyde groups in a cyclic arrangement. In their absence, only a polymeric material was observed [13]. A similar observation was made with the reaction of a diamine with an activated diacid. Hydrogen bonding has also directed dimerization by olefin metathesis [14].

Preorganization based on a network of hydrogen bonds appears to be relevant in our 24-atom macrocycles (Figure 1). Treatment of the bifunctional monomers with trifluoroacetic acid leads to quantitative dimerization. Upon the acid-catalyzed deprotection, the hydrazine reacts with the acetal to yield the first hydrazone, our so-called “hypothetical preorganized intermediate”. We posited that protonation of the triazines contributes to a network of hydrogen bonds that templates the cyclization reaction by localizing the acetal and hydrazine groups to promote dimer formation. Experimental and computational evidence both supported this thesis. Indeed, the lack of evidence for oligomeric or polymeric species was surprising because such products are commonly observed in dynamic covalent chemistry [15,16,17,18,19].

Experimental evidence for templating derived from (1) the quantitative nature of the reaction across a variety of substrates, (2) the observation of the proton template and (3) network of hydrogen bonds in the products, and (4) the reproduction of the templating effect within acyclic models. Computation provided (1) an evaluation of ground-state energies of isomeric models, both neutral and protonated, that explained the site of protonation and shift in isomer distributions and (2) the energetic contribution of hydrogen bonding to stability. Each topic is elaborated upon in the following paragraphs. The molecules probed in this study appear in [Fig molecules-28-01144-ch001].

## 2. Results and Discussion

Macrocyclization is quantitative. The current method employed for macrocyclization proceeded in quantitative yields to the limits of detection by NMR spectroscopy (>95%). That is, upon evaporation of the solvent, no workup was necessary. However, even before the adoption of the current method, the yields for cyclization were high. In the initial report, macrocycles were subjected to purification by centrifugation and chromatography [20]. Yields averaged 90%. In a second study of the ring size, centrifugation yielded a pellet that was washed with dichloromethane [21]. Yields ranged from 83%–100%. The observation of a preponderance of dimer over oligomeric or polymeric products was consistent with templating.

Protonation of the triazine ring was consistently observed in the macrocyclic products. The reaction conditions employed for macrocyclization, 50% trifluoroacetic acid in dichloromethane, provided the source of protons as well as the trifluoroacetate counterion. Protonation of the triazine ring was consistently observed in the macrocycles prepared to date in both solution and the solid state. Protonation occurred at the site opposite the auxiliary group, dimethylamine or morpholine, as illustrated in Figure 1. Figure 1 shows the crystal structure of a valine macrocycle. The proton was not assigned. It was identified based on the observed electron density.

The hydrogen bonding network was present in the macrocycle. Figure 1 also reveals the network of hydrogen bonds in the products matched that proposed in the preorganized intermediate. The amide carbonyl of one subunit engaged in hydrogen bonding with the other subunit. The carbonyl formed two hydrogen bonds, one with the protonated triazine and the other with α-NH of the amino acid.

Hydrogen bonding was also apparent in the ^1^H NMR spectra of these macrocycles. Figure 2 shows the appearance of well-resolved resonances in the downfield “fingerprint” region of the NMR spectrum. Both the chemical shift and shape were indicative of hydrogen bonding interactions. While the intermediates showed numerous broad resonances between 6 and 9 ppm, these shifted to sharp signals appearing between 7 and 13 ppm for the macrocycles [22]. Analysis of 2-D ROESY spectra established that the solution structure matched that observed in the solid state. The threonine and isoleucine macrocycles also adopted the same folded conformation in solution with the same intramolecular network of hydrogen bonds.

Rotamers were observed in the NMR spectra of intermediates but not in the macrocycle. Substituted triazines existed in an equilibrium of rotational isomers, the so-called rotamers, due to a hindered rotation about the triazine-*N* bond [23,24,25]. Experimental and computational models placed the barrier of rotation at 15–16 kcal/mol [23]. These rotamers are illustrated in [Fig molecules-28-01144-ch002] using an ethylamide intermediate. The geometry is indicated by the naming convention (I–IV), which is conserved throughout this report. Evidence for rotamers appeared in the ^1^H NMR spectrum of the carboxylic acid intermediates and the monomer, manifesting as multiple resonances in the NMR spectrum (see the Appendix A for examples). For example, the α-proton of the amino acid often produced three of the four independent resonances at ~4.5 ppm, each corresponding to a different rotamer. The exchangeable NH protons region between 6 ppm and 9 ppm showed a large number of resonances as well.

In stark contrast, however, only a single rotamer, I, was observed in the macrocycles. This situation could not be dismissed readily as the result of constraints imposed by cyclization. Computational models using well-tempered metadynamics suggested that these macrocycles can adopt multiple conformations [26]. The physical manipulation of Corey–Pauling–Koltun models corroborated these conclusions. Reports of the use of substituted triazines, melamines, in molecular recognition led to a second explanation: the rotamer equilibrium can be influenced by noncovalent interactions. From Figure 2, it is evident that protonation was sufficient to bias the rotamer equilibrium. Computation offered insight into these energetic costs.

Computational models of ground-state energies showed that the rotamers were isoenergetic when neutral. DFT calculations were used to predict relative ground-state energies of models, shown in Figure 3. These models were intended to reproduce the dimethylamine auxiliary group (with fidelity), the amino acid (with methylamine that apes the α-NH and α-carbon), and the hydrazone (limited to the hydrazone of acetaldehyde). The negligible differences in the ground-state energies of the rotamers were consistent with the expectation that bias to one species could be realized with weak, noncovalent interactions—including one or more hydrogen bonds.

Computational models revealed energetic preferences for specific rotamers upon protonation. To elaborate the models to account for protonation, isodesmic reactions were used, as illustrated in Figure 2.

Figure 4 shows the relative energy values (in kcal/mol) for protonation at different sites for each of the four rotamers of the model. Only the trans-hydrazone was modeled. The *cis*-hydrazone was not observed experimentally. To validate the computational model, we analyzed the protonation events at the possible sites of triaminotriazines (melamines) bearing from 1 to 6 *N*-methyl groups and compared the predicted energies to measured pKa values [27,28].

The data revealed a few trends. First, protonation of the hydrazone was significantly disfavored (red) over other sites, with an average energy of 9.7 kcal/mol above the minimum. Second, the most favorable site of protonation (blue) was on a triazine nitrogen adjacent to the hydrazone and oriented such that the lone pair of the hydrazone was available to contribute electron density. The calculated energies for these sites across the rotamer population ranged from the 0.0 to 0.9 kcal/mol above the minimum. We considered these rotamers to be isoenergetic at this level of theory. This site agreed with chemical intuition and with the solution-state and solid-state structures as well. The site opposite the hydrazone (green) also represented a favorable site for protonation, with an average energy of 1.6 kcal/mol above the minimum. The final site for protonation (orange) put the proton in proximity of the hydrazone but without the advantage of the hydrazone’s lone pair of electrons. These energies ranged from 2.7–3.2 kcal/mol above the minimum, a difference that was significant compared to the lowest-energy sites (blue).

Experimental data derived from solution and solid-phase structure analyses of macrocycles using NMR and X-ray crystallography, respectively, showed that only rotamer I was observed. This protonation event occurred at 0.9 kcal/mol higher energy than the minimum. Given the negligible energetic penalty, we turned our attention to whether the hydrogen bonding network provided a greater amount of stabilizing energy.

Increasing the number of hydrogen bonds increasingly stabilized the preorganized intermediate. Using M06-2X/6-31+G(d) DFT calculations and a continuum model for solvent, several optimizations of the extended monohydrazone converged to partially folded states. Figure 5 shows representative structures. The low-energy structure of the neutral intermediate was stabilized by a single hydrogen bond, conveying a 6.4 kcal/mol benefit over the extended state. Enthalpic penalties appeared to preclude the formation of two hydrogen bonds in the lowest-energy structure.

Two different monoprotonated intermediates were modeled. Both contributed more stabilizing energy than the neutral molecule. Protonating the distal triazine afforded two hydrogen bonds and 10.2 kcal/mol of stabilization. Protonating the triazine closer to the hydrazone afforded two hydrogen bonds and 12.4 kcal/mol of stabilization. The doubly protonated intermediate yielded four hydrogen bonds and the greatest stabilization energy, 14.2 kcal/mol. A plot of folding energy vs. number of protons was reasonably linear (R^2^ = 0.90) with a slope of 3.9 kcal/mol of folding energy for each proton added. The data supported the belief that templating and dimer selectivity should be favored at low pH.

Computational models of the products revealed significant stabilization energy from hydrogen bonding. Solution-phase studies of these macrocycles corroborated the hydrogen bonding network illustrated in Figure 1. To determine the energetic contributions of these hydrogen bonds in the product macrocycles, we modeled neutral, monoprotonated, and doubly protonated macrocycles using a solvent continuum. To simplify calculations, the sidechain was truncated to a single methyl group (an alanine equivalent). The calculated structures are shown in Figure 6.

In contrast to the glycine macrocycle, which oriented the carbonyl groups outward (out) to form bridging contacts between unit cells in the crystal structure, the valine and isoleucine macrocycles oriented the carbonyl groups inward (in) to form four productive hydrogen bonding interactions with the protonated triazines. In solution, all macrocycles favored the inward conformation. In the neutral molecule, orienting the carbonyls inward created unfavorable interactions, leading to a penalty of 4.7 kcal/mol when compared with the outward orientation. When doubly protonated, however, an inward orientation of the carbonyls was favored over the outward orientation by 8.7 kcal/mol. The energy difference between the carbonyl orientations for a monoprotonated macrocycle was within 1 kcal/mol at this level of theory.

Modeling noncovalent interactions in acyclic systems supported templating and preorganization. To conclude these studies, we looked to recapitulate the preorganization experimentally in acyclic systems. The model design was guided by crystal structures and intuition. If preorganization was important, we hypothesized that the rotamer equilibrium would shift to the rotamer that facilitated the formation of the hydrogen bond. The NMR spectra of the ethylamide model systems substantiated this belief (Figure 2). In all cases, the models favored one rotamer and the spectra were well resolved, a stark contrast to the spectra of the precursors (available in the Appendix A).

The spectra also revealed some global trends. First, and indicated with a green asterisk, the model compounds showed markedly similar chemical shifts across a range of resonances. This behavior was distinct from that of the macrocycles that subdivided into two populations, indicated with the yellow- and blue-shaded regions of the spectra. Second, as indicated by the arrows, similar chemical shift differences were observed between the models and the corresponding macrocycles. These changes were pronounced for C-NH and α-NH, which engaged in hydrogen bonding interactions either indirectly or directly. The negligible changes in the chemical shifts of H+ and NNH suggested that those environments were very similar in the model and macrocycle.

## 3. Experimental

Computational methods. The calculations used the Gaussian 16 electronic structure package [29]. The calculations used density functional theory (DFT) [30] at the M062X/6-311++(2d,2p) [31,32,33] level using the solvation model based on density (SMD) to model the water solvent [34]. The calculations used ωB97X-D/6-31+G(d,p) DFT. All geometries were optimized in SMD solvent. The pKa calculations were validated against a set of experimental melamine derivatives pKa [27]. Additional details are available in the Appendix A.

Specific experimental methods and synthesis. The details of the synthesis of these compounds and the characterization data can be found in the Appendix A.

NMR Spectroscopy. Room temperature ^1^H NMR spectra were recorded on a 400 MHz Bruker Avance spectrometer. Chemical shifts for the ^1^H NMR spectra (in parts per million) were referenced to a corresponding solvent resonance (e.g., DMSO-d_6_, δ = 2.52 ppm). The ^13^C{^1^H} NMR spectra were recorded on the same 400 MHz Bruker spectrometer referenced to the corresponding solvent resonance. All 2D spectra were taken on the 400 MHz Bruker Avance relative to corresponding solvent resonances. Low-temperature spectra were acquired on a 500 MHz Varian NMR spectrometer at the University of North Texas in Denton. The identification of NMR signals were as follows: s = singlet, d = doublet, t = triplet, dd = doublet of doublet, m = multiplet, etc. NMR solvents were deuterated and purchased as a bottle or ampule. Structural assignments were made with and supported by two-dimensional NMR experiments (COSY, HSQC, and rOesy).

General chemistry. Flash chromatography experiments were carried out on silica gel with a porosity of 60Å, particle size of 50–63 m, surface area of 500–600 m^2^/g, a bulk density of 0.4 g/mL, and a pH range of 6.5–7.5. Dichloromethane/methanol was used as the eluent for chromatographic purification. Thin-layer chromatography experiments were carried out in sealed chambers and visualized with UV or submersion in ninhydrin (1.5 g ninhydrin in 100 mL of *n*-butanol and 3.0 mL acetic acid) followed by heating. Excess solvents were removed via rotary evaporation on a Buchi Rotavapor RII with a Welch Self-Cleaning Dry Vacuum System. All workup and purification procedures were carried out with reagent-grade solvents under ambient atmosphere.

## 4. Conclusions

Experimental and computational evidence support a model for templated cyclization. Computation suggests that hydrogen bonding contributes 14.2 kcal/mol of energy to the preorganized intermediate when compared with the extended model. This network of hydrogen bonds is preserved in the structures of the products. The magnitude of the contribution is consistent with the observed quantitative yields. Furthering the argument for preorganization, protonation reduces the number of rotamers sampled in the acyclic models. This preference is unsurprising given that the cost for adopting the desired rotamer in the protonated state was negligible in comparison.

The model that emerges to anchor our intuition, as advanced in Figure 1, is validated. That is, upon condensation to form the first hydrazone, a hydrogen bonding network forms within the molecule. The entropic costs predicted to be associated with this conformational change are hypothesized to be offset by strong hydrogen bonds. These quantitative yields appear to be independent of the choice of the amino acid. Further studies will evaluate the generality of this prediction.

## Data Availability

Not applicable.

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
