# Peer review of "Computational and Experimental Evidence for Templated Macrocyclization: The Role of a Hydrogen Bond Network in the Quantitative Dimerization of 24-Atom Macrocycles"

_molecules, 2023, doi:10.3390/molecules28031144_

Round 1

Reviewer 1 Report

The work is focused on discovering of the influence of so-called 'weak interactions' on the formation of new molecules or supramolecular aggregates. Nowadays  that is really actual and actively developing area of chemistry. So, the work is of high novelty and significance. The work is well done, all conclusions a proved by experimental data and calculations. Maybe the paper will be interesting for not very broad scientific community, but it will exactly find its audience. I do suggest to accept the paper for publishing as it is. Probably some language corrections are needed.

Author Response

Thank you for the supportive response to our work.  We hope that you are right in believing it will find a positive reception with the audience.  

Reviewer 2 Report

Dear Authors,

The manuscript “Computational and Experimental Evidence for Templated Macrocyclization: The Role of a Hydrogen Bond Network in the Quantitative Dimerization of 24-Atom Macrocycles” is a very interesting article that shows a nice example of how the supramolecular preorganization of molecules, through hydrogen bond networks, can be exploited to achieve high yields in the synthesis of macrocycles, which is still a recurrent problem in organic chemistry.

The article is very well written, and the hypothesis is well supported by the presented evidence, acquired by a combination of experimental techniques and theoretical tools.

I just want to suggest some very minor changes in the manuscript, for the authors to consider:

-     Line 37. Reference 4 is repeated.

-      I would suggest the authors to homogenize the size and resolution of the images throughout the manuscript.

-       I think the experimental details for the NMR and the General Chemistry, which are now in the ESI, should be in the Experimental section of the main manuscript, since they are relevant for understanding the results presented there.

-   The numbering of the References is duplicated.

Author Response

Thank you for the positive endorsement of our work.  We have moved the experimental details from the SI to the body of the manuscript as requested, corrected the referencing errors and readjusted the scale of the figures.